# Analysis of the Influence of Socio-Demographic Variables and Some Nutrition and Lifestyle Habits on Beverage Consumption in the Spanish Population

**DOI:** 10.3390/foods12234310

**Published:** 2023-11-29

**Authors:** Elena Sandri, Vicent Modesto i Alapont, Eva Cantín Larumbe, Germán Cerdá Olmedo

**Affiliations:** 1Faculty of Medicine and Health Sciences, Catholic University of Valencia San Vicente Mártir, c/Quevedo 2, 46001 Valencia, Spain; german.cerda@ucv.es; 2Doctoral School, Catholic University of Valencia San Vicente Mártir, c/Quevedo 2, 46001 Valencia, Spain; 3Hospital Universitario y Politécnico La Fe de Valencia, Avenida de Fernando Abril Martorell 106, 46017 Valencia, Spain; vicent.modesto@gmail.com; 4Escuela Técnica Superior de Ingeniería Informática, Polytechnical University of Valencia, Camí de Vera s/n, 46022 Valencia, Spain; ecanlar@etsinf.upv.es

**Keywords:** beverage, Healthy Eating Index, Spanish population, socio-economic factors, lifestyle habits

## Abstract

Beverages and drinks play a significant role in maintaining the integral health of individuals. The aim of this study is to discover the pattern of beverage consumption in different groups of the Spanish population and to investigate its relationship with other nutritional variables and habits. To achieve the objectives, an observational, descriptive and cross-sectional study was conducted. For data collection, a questionnaire was designed and validated that explored different beverage and food consumption variables as well as socio-demographic and lifestyle variables. The instrument was disseminated, among the Spanish young adult population, through snowball sampling using social networks, collecting a sample of 17,541 valid surveys. Bivariate comparative analyses and correlation analyses were performed, and finally, the principal component analysis (PCA) method was used in order to study the relationships between variables related to drinking and health. The main results show significant differences in the pattern of beverage consumption between the socio-demographic variables of sex, age and educational level, as well as between different areas of Spain, while the PCA model shows the relationship between the consumption of sugar-sweetened beverages with the Healthy Nutrition Index of the population and sport practice. Based on the results of the study, the following conclusions were reached: the beverage consumption pattern of the Spanish population is affected by socio-demographic variables. Healthier drinking habits affect the nutrition and health of the population.

## 1. Introduction

Beverages and drinks are essential elements in maintaining the health and well-being of individuals [1]. Staying adequately hydrated is essential for maintaining bodily functions such as digestion, circulation, temperature regulation, and the transport of nutrients and oxygen [2,3], and the type of beverages consumed can have a significant impact on overall health [4,5,6]. Water is the most consumed drink and the primary source of hydration, but other beverages also contribute to fluid intake.

Body hydration is not the only health contribution that beverages make, many beverages, such as fruit juices and smoothies, can provide essential vitamins, minerals, and antioxidants [7,8] that support the body’s immune system and overall vitality. Drinks like milk provide calcium and vitamin D [9], which are crucial for maintaining strong and healthy bones, and certain beverages, like herbal teas, can aid in digestion and soothe gastrointestinal discomfort [10,11]. Certain beverages, like green tea and red wine (in moderation), contain antioxidants that may support heart health [12,13]. Finally, beverages that contain carbohydrates, such as sports drinks, or drinks containing excitants, such as coffee, can provide a quick source of energy during physical activities or during the working day [14,15].

However, it is important to be mindful of the type and quantity of beverages consumed. Some beverages can be high in added sugars, artificial flavours, and empty calories, which may contribute to health issues such as obesity, diabetes, and dental problems [4,16,17]. Choosing low-calorie, nutrient-dense beverages can help with weight management and control calorie intake [1,18]. It is advisable to prioritise the consumption of water relative to that of other beverages to keep the body hydrated, or choose beverages that provide nutritional benefits without excessive sugar or unhealthy additives [19].

The types of drinks people consume are often influenced by cultural and socioeconomic factors. These factors play a significant role in shaping dietary preferences and habits, including beverage choices. Another important factor that can influence the choice of one beverage over another is the availability of ingredients and the climatic conditions of the place where you live [20]. Or cultural norms can influence whether certain beverages are considered acceptable or appropriate to consume in specific settings. 

If we analyse the influence of the economic or cultural level, it is observed that people with higher incomes can afford to choose from a wider range of beverages and may be more likely to afford premium or specialty drinks [21], or that people with a higher level of education and culture may be better informed about healthier drink options [22].

The impact that beverage consumption can have on a person’s health is significant, which is why it is important to understand the beverage consumption patterns of a population and to investigate how these relate to socio-cultural factors as well as to other nutritional habits and lifestyles. Understanding these influences is important for public health efforts, as it helps in developing strategies to promote healthier beverage choices that are culturally sensitive and accessible to people from different socioeconomic backgrounds.

Despite the growing concern about population nutrition, there are still few studies that focus specifically on the impact of beverages on nutrition and health, and that analyse the contribution of different types of beverages.

While there are numerous studies exploring beverage consumption, most of them focus only on measuring the health impact of consuming a specific beverage. There are, for example, studies conducted on energy drinks in Denmark [23] or the United States [24,25], studies focusing on sugary drinks such as those conducted in the UK [26], Peru [27], India [28] or Sweden [29], or studies focused on exploring the consumption of carbonated beverages [30]. In contrast, there are few studies that focus on exploring the demographic determinants of beverage consumption in general. In this respect, the study conducted on the French population in 2016, which includes a very large sample, is worth mentioning [31].

With regard to Spain, studies have been carried out analysing beverage consumption in specific regions of the territory, such as Catalonia [32], the Balearic Islands [33] or in the Community of Madrid [34]. Furthermore, there are other studies that focus only on the consumption of specific beverages such as energy drinks [35] or alcohol [36,37,38,39] that have been studied in depth. Spanish studies analysing the impact of different types of beverages on health and their socio-demographic determinants are scarce and the sample used is not very large. The ANIBES study [40] has a sample of 2285 participants, while the study conducted by Cíntia Ferreira-Pêgo et al. [41] has a sample of 1262 adults.

This study, on the beverage consumption of the Spanish population, aims to contribute to the evidence in this field. The large number of participants is undoubtedly one of its strengths and may allow extrapolable results to be obtained.

The objectives of this study are as follows: To explore the associations between beverage consumption and sociodemographic factors. Specifically, the relationship between age, sex, educational level, purchasing power, place and type of residence of the person and beverage consumption was analysed.To analyse the relationship between the frequency of consumption of a given beverage and some nutritional, lifestyle and health variables.

Specifically, the variables explored were as follows: the Healthy Nutrition Index, body mass index, self-perceived level of health, sedentary lifestyle, physical activity, and time and quality of sleep.

## 2. Materials and Methods

### 2.1. Study and Sampling

A prospective cross-sectional study was carried out on the Spanish population aged between 18 and 45 years living in Spain. We decided to limit the sample to those ages to minimise the possible bias caused by the greater difficulty in accessing the internet and social networks that most of the older population has. Subjects were excluded if, at the time of answering the survey, they had any pathology or limitation that could affect their diet, such as hospitalisation or confinement.

### 2.2. Ethical Approval

The study was carried out with the approval of the ethics committee of the Universidad Catolica de Valencia (UCV/2019-2020/152). All data collected came with the informed consent of the participants in the study, and the ethical recommendations contained in the Declaration of Helsinki [42] were followed at all times.

### 2.3. Instrument

The instrument used for data collection was a self-developed questionnaire consisting of several parts: the first part collected information on anthropometric data such as weight, height and socio-demographic variables, the second part collected information on different food groups and their respective frequencies of consumption, the third part collected information on drinking habits, and the fourth and last part collected data on sports and lifestyle habits related to health. Rigorous methodological approaches were followed for instrument development and psychometric testing.

The questionnaire was validated through a pilot group of 53 people, with similar characteristics to the study population, and through a nominal group of 7 health experts, namely a nutritionist, two family doctors, two psychologists, a social educator and an expert from the world of communication.

### 2.4. Data Collection

The questionnaire was disseminated using non-probabilistic snowball sampling. The support used was the Google Form program and the channels used were mostly telematic; firstly, the Instagram account @elretonutricional was used, but so were the researchers’ personal social networks and mailings to various associations.

On the other hand, we also tried to disseminate the questionnaire physically, in order to facilitate access to those people who are less fond of networks and technology, using posters hung in different shops and businesses where many different people go every day. The data collection period was from August 2020 to November 2021.

### 2.5. Variables

The socio-demographic variables used were as follows: sex (which was analysed in binary form, distinguishing between male and female), age (which was categorised between young people who were (18–30) years old and adults who were (31–45) years old), level of studies (which was categorised into basic studies (no studies, only primary or secondary studies, vocational training or baccalaureate) and higher studies (degree, master’s and PhD)), type of residence (which was categorised into those living in the family home and those living outside), shared residence (where a distinction was made between those living alone and those living with others) and place of residence (where a distinction was made between the different regions of Spain).

To analyse nutrition, we decided to use the IASE index (Índice de Alimentación Saludable para la población Española) [43] by adapting a reduced version of it. The adapted index reports a maximum score of 73 points, in accordance with the degree of compliance with the recommendations proposed by the Spanish Society of Community Nutrition (SENC) [44]. According to the IASE score obtained, the nutrition habits of the population can be classified into 3 categories: 58.4 < IASE < 73: “Healthy”; 36.5 < IASE < 58.4: “Needs changes”; IASE < 36.5: “Unhealthy”. Table 1 shows the categorisation of variables used for the IASE.

The beverage consumption variables were as follows: water, sugary soft drinks, juice, coffee and energy drinks, drinks at mealtimes, alcohol and binge drinking. The health and lifestyle variables were a sedentary lifestyle, sleeping hours and sleep quality. and were categorised with a score from 1 to 4 according to the indicated criteria in Table 2, except for body mass index (BMI) and minutes of sport, which was used as a numerical.

### 2.6. Statistical Analysis

To begin with, the data were cleaned by eliminating erroneous data, as a consequence of incorrect entry by the respondent, or atypical data on both height and weight, trying to eliminate BMI values below 14 and above 40. Once the information had been cleaned, the corresponding descriptive and inferential statistical analysis was carried out using Excel 2013 for Windows and SPSS v.23 (SPSS Inc., Chicago, IL, USA). 

The non-parametric Mann–Whitney U test for independent samples (to analyse dichotomous variables) and the Kruskal–Wallis test (for comparison between more than two groups) were used to compare means and study significance. The choice was motivated by the fact that the normality criterion had previously been tested using the Kolmogorov–Smirnov test with Lilliefors correction and the results had indicated that the data did not follow a normal distribution. Both tests were applied with a significance level of 0.05.

Performing an analysis for each pair of variables studied is not functional, so we decided to use the principal component analysis (PCA) method to explore the relationships between the ordinal numerical variables, since this method allows the dimensionality of a model to be reduced. Two principal components were chosen that explain a percentage of variability of about 24%, as can be seen in Figure 1.

The first dimension (or principal component) is mainly explained by sugary drinks, IASE, meal beverages, and sleep quality, as can be seen in Figure 2. Figure 3 shows how the variables alcohol and getting drunk are the main contributors to the second dimension. This method was carried out with RStudio 4.3.0.

## 3. Results

Table 3 shows the socio-demographic characteristics of the respondents. After data collection and subsequent filtering, we finally obtained 17,541 valid surveys representative of people resident in the whole of Spain, including the islands.

The behaviour of the sample was studied in relation to the frequency of consumption of water, sugary soft drinks, juices, coffee and energy drinks, and alcohol, and the frequency of binge drinking, distinguishing this via the socio-demographic variables of sex, age, educational level, type of residence and shared residence (Table 4).

Significant differences in the frequency of consumption of the same measures were also analysed with respect to the region of residence in Spain (Table 5).

As all *p*-values for each Kruskal–Wallis between the drinking variables and the autonomous communities of Spain (regions of Spain) are significant, it could be interesting to see the differences in a map (Figure 4) particularly with respect to sugar-sweetened beverages, since, together with alcohol, they are the ones that have the greatest impact on health. 

It can be observed how the consumption of sugar-sweetened beverages is higher in the northern part of Spain than in the centre-south, particularly in the autonomous communities of Cantabria and Asturias, followed by Galicia and the Balearic Islands.

Before applying a PCA model, it is essential to study the correlation between the variables implied in the model. A Pearson correlation graph can be seen in Figure 5. It can be observed that there are no high correlations between the variables, except for get drunk and alcohol (0.295), as is logical, sleeping hours and sleeping quality (0.366), as is logical too, and more interestingly between sugary drinks and the IASE (0.365). 

People who do not drink alcohol or drink alcohol infrequently (high punctuation in alcohol) tend not to get drunk (high punctuation in get drunk). Furthermore, survey respondents who do not usually drink sugary drinks also have a good IASE. Finally, those who have good sleep quality sleep more hours. However, it is worth noting that all these correlations are not high.

Figure 6 shows the PCA model and helps to explore the relationships between all variables related to drinks and those related with health.

In the PCA, via the colour and length of the arrows, following the colour scale that marks the level of contribution of the variables, it can be seen how the variables sedentary lifestyle, and frequency of consumption of juice, coffee and energy drinks have the least influence on the model; in fact, they are coloured in blue and their lengths are very short. The variables educational level, water consumption, sport and BMI, which are coloured green, also have a limited contribution, while the variables related to sleep, binge drinking and IASE have a medium contribution, with their arrow being coloured in yellow and of considerable length. Finally, the consumption of alcohol and that of sugar-sweetened beverages are the variables with the greatest influence on health, with their arrows being the longest and coloured orange-red in the PCA model.

Looking now at the orientation of the arrows, alcohol is more related to get drunk and coffee and energy drinks. However, this last variable contributes less to the model. Additionally, sleeping hours and sleep quality are highly positively correlated and negatively correlated with BMI. Therefore, the more time spent and the greater quality of sleep, the smaller the punctuation in BMI. Finally, as already observed in Pearson’s correlation model, IASE and sugary drinks are correlated. Then, those who drink less sugary drinks have a higher IASE.

## 4. Discussion

One of the main findings of this study is that, for the young adult Spanish population, the frequency of beverage consumption shows a clear dependence on socio-demographic variables, particularly with regard to gender, age and educational attainment (see Table 4 and Appendix A). This is in line with previous studies in the literature that focus on specific beverages in different regions of the world, such as the one conducted by Jozaa Z AlTamimi [29], the other one conducted by Manu Raj Mathur [45], both focusing on sugar-sweetened beverages, several that focus on alcohol consumption [46,47,48], or those conducted on the consumption of energy drinks [23,24] or caffeinated beverages [49]. This dependence of beverage consumption on socio-demographic factors is of fundamental importance in public health. Knowing that gender, age, level of education or place of residence determines people’s choice or prioritisation of one drink or another can help to design and disseminate more targeted and therefore more effective public health actions. Training or awareness-raising campaigns can be targeted at a particular gender or age group, or personalised according to the place of residence and therefore the climatic conditions or customs of that territory. We can also use a simpler and plainer message for those with basic education and more scientific and specific language for those with higher education. Although it is certainly necessary to carry out more studies focused on specific socio-demographic variables, which will allow a further refinement of the results, so that the public health measures adopted can be effective, the results obtained can be a starting point.

On the other hand, the data allow us to observe that living alone as opposed to living with other people, or living in a family home as opposed to living away from home, although there are significant differences between categories in beverage consumption, tare not as marked as in the case of other socio-demographic variables (see Figure A4 and Figure A5 of Appendix A). This is important given the steady increase in recent years in Western society in people living alone [50,51,52]. While it has been observed that living outside or away from the family home influences some nutritional patterns and health habits [53,54,55,56], the results seem to indicate that it has little influence on drinking. Perhaps it could be interesting to study, and therefore make a future line of work, whether this behaviour, observed for the young adult population, also holds for the older population, which is generally a population more at risk of suffering the detrimental effects of loneliness and living alone [57,58,59,60]. Our research group, without going any further, plans to carry out a new round of dissemination of the questionnaire and a new data collection round, this time using the inclusion criterion of being over 45 years old. In this way, we will be able to obtain results from adults and elderly people that can be compared with those obtained from the younger age group.

Another result that may attract attention is the consumption of sugar-sweetened beverages according to the regions of Spain; although the differences in absolute values are not very marked, they are statistically significant. Figure 4 visually shows how it is higher in general in the northern part of Spain than in the centre-south, particularly in the autonomous communities of Cantabria and Asturias, followed by Galicia and the Balearic Islands. This could be explained by the fact that these regions have different cultural and gastronomic aspects compared to those in other parts of Spain, and this may also have a significant influence on the type of beverages consumed in that area. The relationship between culture, food and beverages is intrinsic and complex, and may be a reflection of history, climate, availability of ingredients and local traditions [61,62]. It could be interesting in the future to carry out studies on food, nutrition and beverage consumption habits focused on specific regions of the Spanish territory in order to study in more detail the influence of gastronomy and customs and how they vary at the territorial level. Such studies, in a specific area, could perhaps use mixed techniques, both quantitative and qualitative. They could combine the dissemination of a questionnaire to the population, with experimental data collection in restaurants and establishments to analyse the type of diets and drinks offered and the consumption pattern of customers. They could also use the data collected in national surveys to analyse the food purchasing pattern of the population.

Another striking result of the study is the relationship between the practice of physical activity and the consumption of sugar-sweetened beverages, which can be seen visually in the PCA graph in Figure 6. In this figure, it can be seen how the lines representing the contribution of the two variables are almost parallel and point in the same direction. Increased physical activity increases the need for hydration due to the loss of fluids and electrolytes through sweat [63,64,65]. The most common beverages for hydration during exercise are typically water, sports drinks designed to replenish electrolytes and, in some cases, carbohydrate drinks to provide additional energy. The choice of beverage depends on the duration and intensity of exercise, as well as personal preference. Sport has become an extremely popular activity in Western society and around the world in recent years. This popularity has led to a significant increase in the movement of money and a growing media interest in sport. All of these factors have led to increasingly aggressive marketing and advertising campaigns by brands seeking to capitalise on this trend. Not all brand marketing and advertising messages provide accurate information about the dietary and nutritional properties of a product and its healthfulness [66,67,68]. While sugary drinks can be very appealing in taste, and in some cases effective thirst quenchers, they are not always the healthiest option for replenishing fluids after sporting activities [69,70]. This opens up another important field of public health, which especially affects the young and young adult population, which is the one that generally practices sports on a more consistent basis. It is a priority to design effective training campaigns on the consumption of beverages before, during and after sports practice, to teach the population to practice healthy hydration and to choose products that are not only commercially attractive but also nutritionally adequate. However, in order to achieve this effectively, it will be necessary to rely on successive studies specifically designed to analyse the consumption of sugar-sweetened beverages in young people, sportsmen and sportswomen.

### Strengths and Limitations

One of the weaknesses of the study is represented by the sampling technique used. Snowball sampling was used, which offers the advantage of reaching a large number of responses, but may result in a biased sample because the sample collected depends on participants recommending other potential participants.

Analysing our sample, we detected some self-selection bias in that the sample collected had a large number of people concerned about nutrition and health. This is mainly due to the fact that when the survey was distributed through social networks, several influencers and professionals working in these fields supported the dissemination of the survey. It can be assumed that the community of followers of these accounts are, for the most part, people interested in nutrition and health who may have higher-than-average knowledge of the subject and who generally seek to adopt healthier habits. 

We tried to minimise self-selection bias by also disseminating the survey physically, outside of social networks. To this end, we contacted different establishments throughout Spain with a heterogeneous public (pharmacies, tobacconists, associations, etc.) and asked them to display a poster in their establishments explaining the objectives of the research and containing a QR code that referred to the survey. In this way, people visiting these establishments could access the survey if they wished to do so.

We should also highlight a gender bias, with 82.26% of those who responded to the survey being female. This result is not surprising because in general there is a greater concern among women about nutrition and health issues, which predisposes them to read and be more interested in these topics and, consequently, to be more inclined to participate in studies in this field.

To minimise, as much as possible, this sex bias, an effort was made to recruit male representatives for the study. Although it was not possible to match female participation, a broadly significant sample of 3111 males was achieved.

Finally, the diffusion of the sample through social networks may induce an age bias in the sample, presenting a higher response rate among the young population. Young people in general are more familiar with and attracted to the use of social networks, tend to devote more time to their use, and were therefore more likely to answer the survey. 

After reflecting on this bias and in order to minimise it, it was finally decided to focus the study on the population between 18 and 45 years of age, ages in which the use of networks is widespread.

The greatest potential of this study undoubtedly lies in the large sample size, which provides great statistical power and reduces the study’s margin of error. Having a large population allows a representation of different demographic groups, which increases the likelihood that the conclusions are representative of the general population. 

Another noteworthy point of the present study is the geographic scope achieved, with a representative sample from every region of Spain, including the islands. Geographic diversity allows a more complete understanding of the subject under investigation and increases the external validity of the study, reducing the possibility of geographic bias.

Finally, the potential of the questionnaire obtained for data collection cannot be overlooked; the ease and speed with which it can be completed makes it suitable for wide dissemination through different channels, thus making it possible to achieve very large and significant samples in different groups.

A next step could be to replicate the research by focusing it on an older target population to explore the behaviour, in terms of beverage consumption and nutrition and lifestyle habits, of adults and the elderly.

## 5. Conclusions

The main results show significant differences in the pattern of beverage consumption between the socio-demographic variables of sex, age and educational level, as well as between different areas of Spain. Furthermore, it can be concluded that living alone or with others and living in the family home or away from home also affect drinking habits, although to a lesser extent than other socio-demographic variables do.

The PCA model shows the inverse relationship between the consumption of sugar-sweetened beverages with the Healthy Nutrition Index of the population and the direct relationship between the consumption of sugar-sweetened beverages and the practice of physical activity. In addition, hours of sleep and sleep quality have been found to correlate with body mass index, leading to the conclusion that those who spend more time sleeping and who have a higher quality of sleep in general have a lower BMI.

It can be concluded that the beverage consumption pattern of the Spanish population is affected by socio-demographic variables. Healthier drinking habits affect the nutrition and health of the population.

## Figures and Tables

**Figure 1 foods-12-04310-f001:**
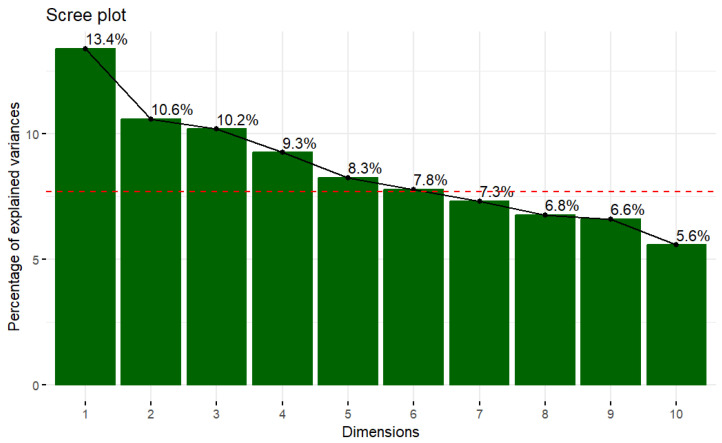
Scree plot of the PCA Model. NOTE: The red line indicates the average percentage of explained variance (if all dimensions are explained as the same).

**Figure 2 foods-12-04310-f002:**
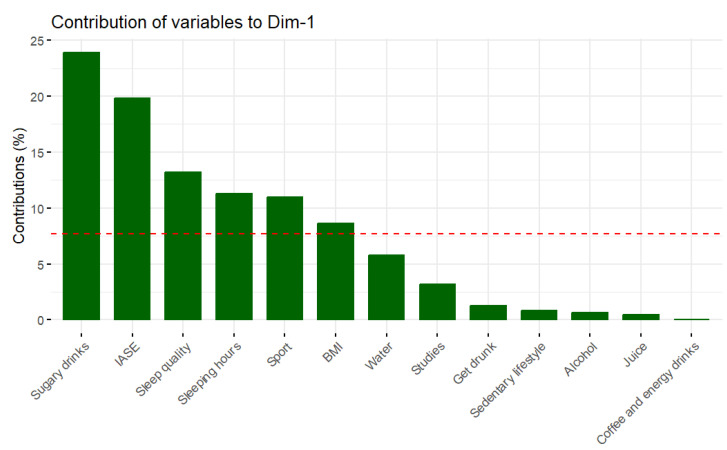
Contribution of variables to the first dimension. NOTE: The red line indicates the average percentage of contribution if all dimensions contribute the same.

**Figure 3 foods-12-04310-f003:**
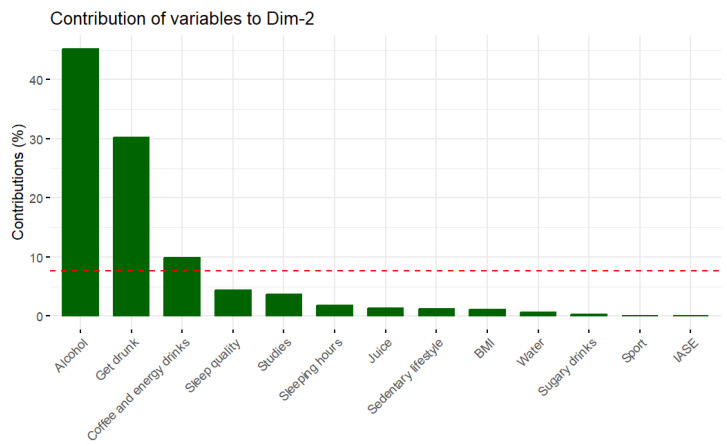
Contribution of variables to the second dimension. NOTE: The red line indicates the average percentage of contribution if all dimensions contribute the same.

**Figure 4 foods-12-04310-f004:**
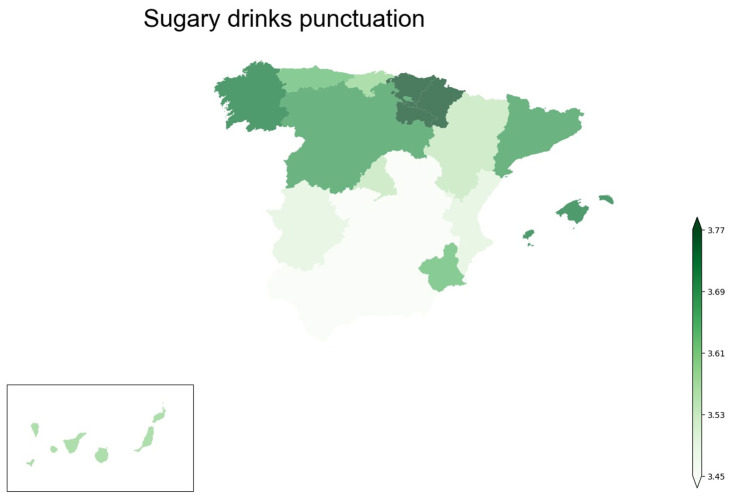
Map of autonomous communities of Spain coloured depending on the punctuation of sugar drinks.

**Figure 5 foods-12-04310-f005:**
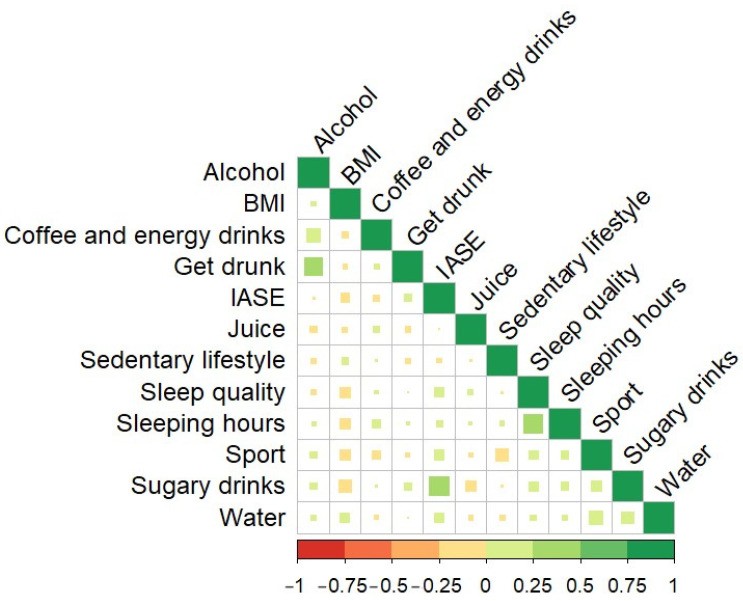
Pearson correlation plot between “the drinking variables” and health variables.

**Figure 6 foods-12-04310-f006:**
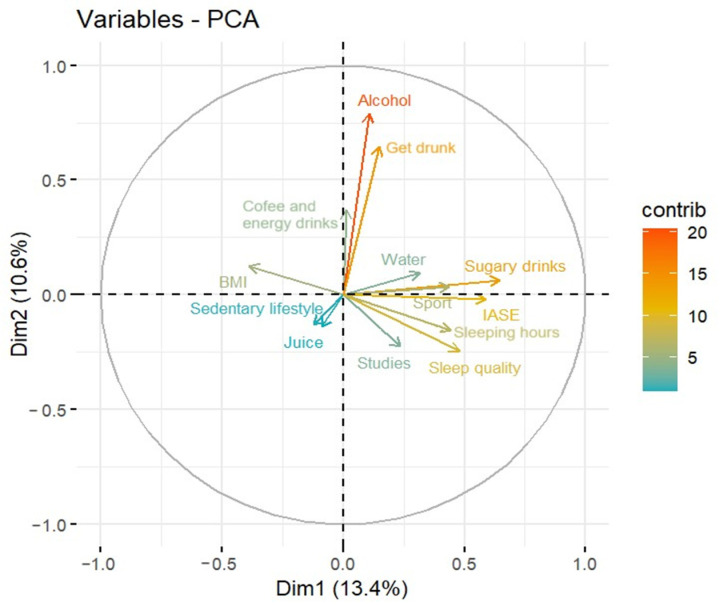
Variables plot of the PCA model.

**Table 1 foods-12-04310-t001:** Conversion table applied to convert qualitative questions into quantitative questions of the nutritional index.

Conversion Table Applied for Iase					
Variables	Score				
	10	7.5	5	2.5	0
Fruit	1 piece/portion per day, 2 to 4 portions per day, 5 or more pieces per day				Never or rarely
Vegetables	Every day	5 or more pieces per week and Between 2 and 4 pieces per week	1 piece/ration per week		Never or rarely
Cereals	Every day	3 or more times a week	1–2 times a week		Never or rarely
Milk	Every day	3 or more times a week	1–2 times a week		Never or rarely
Medium between white and red meat	1–2 times a week	3 or more times a week		Every day	Never or very rarely
Legumes	1–2 times a week	3 or more times a week		Every day	Never or very rarely
Soft drinks	Never or rarely	Very few times (2 times maximum per month)	One glass per week	2 or more glasses per week	Two glasses or less every day and Between 3 and 5 glasses every day and More than 5 glasses every day
Variety	2 points if each of the daily recommendations is met, 1 point if each of the weekly recommendations is met.				

**Table 2 foods-12-04310-t002:** Categorisation of beverage consumption variables and the health and lifestyle variables.

Variable	Category	Score
Sleeping hours	<6 h for night	1
	6 < hours < 7	2
	7 < hours < 8	3
	hours > 8	4
Sleep quality	0 and 1	1
	2	2
	3	3
	4 and 5	4
Water	Never and very rarely (2 max. per month) and 1 glass/cup/week and 2 or more glasses/cups/week	1
	2 glasses/cups or less every day	2
	3 to 5 glasses every day	3
	More than 5 glasses every day	4
Sugary soft drinks, coffee and energy drinks	Never and very rarely (2 glasses max. per month)	4
	One glass per week and 2 or more glasses per week	3
	2 glasses or less every day	2
	3 to 5 glasses and more than 5 glasses every day	1
Juice	Never and very rarely (2 glasses max. per month)	1
	One glass per week and 2 or more glasses per week	2
	2 glasses or less every day	3
	3 to 5 glasses and more than 5 glasses every day	4
Getting drunk	Never or less than once a month	1
	Monthly	2
	Weekly	3
	Daily or almost daily	4
Alcohol consumption	Never or once a month	1
	2–4 times a month	2
	2–3 times a week	3
	4–5 times a week or every day	4
Sedentary lifestyle	Less than 7 h	1
	Between 7 and 9 h	2
	Between 9 and 11 h	3
	More than 11 h	4

**Table 3 foods-12-04310-t003:** Data collected and their socio-demographic characteristics.

		Mean ± SD or N (%)	
Male		3111 (17.74%)	
Female		14,621 (82.26%)	
Age (years)		30.31 ± 7.54	
Male Age (years)		29.98 ± 7.74	
Female Age (years)		30.38 ± 7.49	
	Total	18–13 Years	>30 Years
Age (%N)		9558 (54.49%)	7983 (45.51%)
Male Age (%N)	3111 (17.74%)	1810 (10.32%)	1301 (7.42%)
Female Age (%N)	14,621 (82.26%)	7748 (44.17%)	6682 (38.09%)
Elevel education			
Basic education	5600 (31.93%)	3472 (19.80%)	2128 (12.13%)
Higher education	11,941 (68.07%)	6086 (34.69%)	5855 (33.38%)
Income level			
Low	8207 (46.79%)	4820 (27.48%)	3387 (19.31%)
Medium-high	7785 (44.38%)	3538 (20.17%)	4247 (24.21%)
Don’t know-no answer	1549 (8.83%)	1200 (6.84%)	349 (1.99%)

**Table 4 foods-12-04310-t004:** Comparison of drinking habits differentiated by sociodemographic variables.

	Mean ± SD		
	Male	Female	*p*-Value *
Water	3.54 ± 0.59	3.41 ± 0.63	1.14 × 10^−25^
Sugary drinks	3.52 ± 0.72	3.61 ± 0.65	1.44 × 10^−11^
Juice	1.26 ± 0.56	1.23 ± 0.52	0.0011
Coffee and energy drink	3.23 ± 0.76	3.34 ± 0.68	1.17 × 10^−11^
Alcohol consumption	3.18 ± 0.88	3.33 ± 0.80	3.46 × 10^−16^
Get drunk	3.88 ± 0.40	3.95 ± 0.27	1.16× 10^−27^
	18–30	31–45	*p*-value *
Water	3.45 ± 0.63	3.43 ± 0.62	0.0201
Sugary drinks	3.57 ± 0.67	3.62 ± 0.66	3.98 × 10^−7^
Juice	1.26 ± 0.55	1.19 ± 0.49	5.59 × 10^−23^
Coffee and energy drink	3.41 ± 0.66	3.22 ± 0.72	1.50 × 10^−70^
Alcohol consumption	3.31 ± 0.77	3.29 ± 0.86	0.1654
Get drunk	3.92 ± 0.32	3.95 ± 0.27	2.79 × 10^−15^
	Low studies	High studies	*p*-value *
Water	3.44 ± 0.66	3.44 ± 0.61	0.0504
Sugary drinks	3.50 ± 0.74	3.64 ± 0.62	4.30 × 10^−34^
Juice	1.25 ± 0.54	1.22 ± 0.52	0.0004
Coffee and energy drink	3.37 ± 0.72	3.30 ± 0.69	3.21 × 10^−16^
Alcohol consumption	3.38 ± 0.79	3.26 ± 0.82	1.06 × 10^−23^
Get drunk	3.93 ± 0.32	3.94 ± 0.29	0.0072
	Live away from home	Live at home	*p*-value *
Water	3.45 ± 0.62	3.43 ± 0.63	0.2761
Sugary drinks	3.63 ± 0.64	3.58 ± 0.68	6.18 × 10^−6^
Juice	1.23 ± 0.52	1.23 ± 0.53	0.9380
Coffee and energy drink	3.29 ± 0.69	3.33 ± 0.70	6.21 × 10^−5^
Alcohol consumption	3.18 ± 0.83	3.35 ± 0.80	1.34 × 10^−38^
Get drunk	3.90 ± 0.36	3.95 ± 0.27	7.54 × 10^−20^
	Live alone	Live accompanied	*p*-value *
Water	3.43 ± 0.63	3.47 ± 0.62	0.0095
Sugary drinks	3.59 ± 0.67	3.63 ± 0.66	0.0012
Juice	1.23 ± 0.53	1.22 ± 0.51	0.1592
Coffee and energy drink	3.34 ± 0.69	3.22 ± 0.73	1.50 × 10^−11^
Alcohol consumption	3.31 ± 0.81	3.17 ± 0.85	1..52 × 10^−12^
Get drunk	3.94 ± 0.29	3.90 ± 0.37	6.50 × 10^−7^

* Mann–Whitney test.

**Table 5 foods-12-04310-t005:** Comparison of drinking habits differentiated by autonomous communities of Spain.

	*p*-Value *
Water	9.90 × 10^−13^
Sugary drinks	2.73 × 10^−23^
Juice	2.29 × 10^−11^
Coffee and energy drink	1.21 × 10^−13^
Alcohol consumption	4.64 × 10^−18^
Get drunk	2.94 × 10^−6^

* Kruskal–Wallis test.

## Data Availability

The data presented in this study are available upon request to the corresponding author.

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
