# Peer review of "Analysis of the Influence of Socio-Demographic Variables and Some Nutrition and Lifestyle Habits on Beverage Consumption in the Spanish Population"

_foods, 2023, doi:10.3390/foods12234310_

Round 1

Reviewer 1 Report

Comments and Suggestions for Authors

The study presents a comprehensive analysis of beverage consumption patterns among the young adult Spanish population, providing valuable insights based on socio-demographic variables. While the paper exhibits strengths in its extensive dataset and its regional representation, there are areas that necessitate refinement for enhanced clarity and impact.

Starting with the title, it demands a concise revision to ensure precision and immediate relevance to the core themes of the paper. The abstract, despite being well-articulated, could benefit from a more streamlined structure, removing specific terminologies like "aim," "methods," etc., to make it more impactful and to the point.

The introduction, while establishing a foundation, lacks a clear articulation of the research gap and the specific objectives of the study. It is imperative to delineate these elements distinctly to guide the reader through the research journey and to underscore the study’s significance and novelty.

A dedicated section for literature review would enrich the paper, providing a comprehensive background and situating the study within the broader academic discourse. This would also serve to highlight the unique contribution of the current research.

The methodology section, while sufficiently detailed, could be further strengthened by a thorough discussion on the measurement instruments employed and the data analysis techniques utilized. This would enhance the transparency and replicability of the study, ensuring its robustness.

In terms of implications, limitations, and recommendations, these critical components of a research paper warrant a separate, detailed discussion. Articulating the practical applications of the findings, acknowledging the limitations candidly, and charting a course for future research would solidify the paper’s academic contribution.

The rest of the paper, including the discussion and results sections, demonstrates a well-executed analysis, drawing meaningful connections between beverage consumption and socio-demographic factors. The incorporation of visual aids like tables and figures  has enhance the accessibility of the data, aiding in the reader’s comprehension.

Building upon the initial review, it is evident that with targeted revisions, particularly in the introduction, methodology, and the addition of specific sections for literature review and implications, the paper has the potential to serve as a significant resource in the domain of public health and nutrition. The study's findings are poised to inform targeted and effective public health interventions, tailoring strategies to specific demographic groups for maximized impact.

The study stands as a testament to the intricate relationship between beverage consumption and socio-demographic variables in Spain, offering a springboard for future research and policy-making. With careful refinement, the paper could further solidify its standing and contribute meaningfully to the ongoing discourse in nutrition and public health.

Comments on the Quality of English Language

The language is expected to be more formal in the paper.

Author Response

Dear Reviewer 1

Thank you very much for taking the time to review our article and for the valuable suggestions you have provided. We have tried to work on each of your suggestions in detail and hope that we have improved the quality of the article.

Below are the changes we have made based on each of the suggestions received.

Sincerely,

Elena Sandri, corresponding author.

[email protected]

The study presents a comprehensive analysis of beverage consumption patterns among the young adult Spanish population, providing valuable insights based on socio-demographic variables. While the paper exhibits strengths in its extensive dataset and its regional representation, there are areas that necessitate refinement for enhanced clarity and impact.

Starting with the title, it demands a concise revision to ensure precision and immediate relevance to the core themes of the paper.

Thank you for your suggestion, we propose to modify the title of the article to the following one, which we think better summarizes the objectives of the study

“Analysis of the influence of socio-demographic variables and some nutrition and lifestyle habits on beverage consumption in the Spanish population”.

The abstract, despite being well-articulated, could benefit from a more streamlined structure, removing specific terminologies like "aim," "methods," etc., to make it more impactful and to the point.

Thank you very much for your suggestion, we have modified the abstract by removing the suggested paragraph headings and giving it a unified discursive structure to make it more impactful.

The introduction, while establishing a foundation, lacks a clear articulation of the research gap and the specific objectives of the study. It is imperative to delineate these elements distinctly to guide the reader through the research journey and to underscore the study’s significance and novelty. A dedicated section for literature review would enrich the paper, providing a comprehensive background and situating the study within the broader academic discourse. This would also serve to highlight the unique contribution of the current research.

 Thank you for the suggestion, it has provided us with valid help to refine the wording of the introduction.

Specifically, we have added a bibliographic review part where we have tried to highlight the gaps that this research intends to fill. We have also reworded the paragraph explaining the objectives of the present study to make them clearer and more visible.

The methodology section, while sufficiently detailed, could be further strengthened by a thorough discussion on the measurement instruments employed and the data analysis techniques utilized. This would enhance the transparency and replicability of the study, ensuring its robustness.

Thank you for the suggestion, we have sent the original version of the questionnaire used for data collection, which can be published in annexes, and will certainly help to clarify the measurement instruments used and the reproducibility of the study.

We have also tried to expand the statistical analysis section by trying to give more detail on how the data collected has been treated.

In terms of implications, limitations, and recommendations, these critical components of a research paper warrant a separate, detailed discussion. Articulating the practical applications of the findings, acknowledging the limitations candidly, and charting a course for future research would solidify the paper’s academic contribution.

 Thank you for your suggestion, because of which we have completely rewritten the section on the strengths and limitations of the study, indicating in detail the possible biases of the study and the measures that have been adopted to minimize them, and emphasizing the strengths that this study undoubtedly also presents.

The rest of the paper, including the discussion and results sections, demonstrates a well-executed analysis, drawing meaningful connections between beverage consumption and socio-demographic factors. The incorporation of visual aids like tables and figures has enhance the accessibility of the data, aiding in the reader’s comprehension.

 Building upon the initial review, it is evident that with targeted revisions, particularly in the introduction, methodology, and the addition of specific sections for literature review and implications, the paper has the potential to serve as a significant resource in the domain of public health and nutrition. The study's findings are poised to inform targeted and effective public health interventions, tailoring strategies to specific demographic groups for maximized impact.

 The study stands as a testament to the intricate relationship between beverage consumption and socio-demographic variables in Spain, offering a springboard for future research and policy-making. With careful refinement, the paper could further solidify its standing and contribute meaningfully to the ongoing discourse in nutrition and public health.

Reviewer 2 Report

Comments and Suggestions for Authors

1.Some deficiencies and areas for improvement in the Introduction section could be identified as follows:

(1)The introduction discusses the general importance of beverage consumption and its impacts; however, it does not clearly highlight the specific research gaps that this study intends to address.

(2)The introduction covers a wide range of beverage types, related health impacts, socio-demographic factors, and lifestyle variables. Narrowing the focus to a specific type of beverage or health impact may make this study more scientifically rigorous and manageable.

(3)The introduction could be shorter and more concise, focusing on key aspects that pertain directly to the study. Excessive detail or broad generalizations can dilute the main points.

(4)The Introduction lacks a clear statement regarding the research objectives. It is not clear what the specific predictions or expectations of the research are.

(5)The introduction does not provide a comprehensive review of existing literature on this subject. Although many references have been cited, there is no clear overview of previous findings or theories that could help contextualize the current research.

 2.The Materials and Methods section of this study has several deficiencies and areas that could be improved.

(1)By limiting the sample to those with internet and social network access, and between the ages of 18 and 45, the study introduces a potential sampling bias that may limit the generalizability of results to wider populations.

(2)The use of non-probabilistic snowball sampling may lead to biased results because the sample may not adequately represent the population. Additionally, the use of social networks or personal contacts may introduce selection bias.

(3)The use of a self-developed questionnaire can be a potential source of bias as it may lack standardized metrics or may not have been validated in diverse populations.

 3.The Discussion section of this study had several limitations. The following are some key points to consider:

(1)The discussion suggests the design and dissemination of public health actions based on some of the socio-demographic variables studied. While these results provide a useful starting point, it is important to not overextend the implications of the findings.

(2)While the authors do refer to previous related studies, they could offer more direct comparisons between their own findings and the results of these previous studies to solidify their results, arguments, and conclusions.

(3)Certain aspects, such as local economic conditions, educational facilities, and public health campaigns might have influenced both variables studied (beverage consumption and sociodemographic factors), potentially confounding the results.

(4)While the authors do note the use of an online survey as a limitation, the discussion does not elaborate on other potential limitations such as the potential for recall bias, self-selection bias, non-response bias, or limitations inherent to cross-sectional studies.

(5)The discussion proposes a number of interesting directions for future research but does not provide hypotheses or methods on how this research should be conducted.

4.This conclusion restates the essential findings of this study. Here, the conclusion seems to summarize only a fraction of the study's full results.

Author Response

Dear Reviewer 2

Thank you very much for reviewing our work, it is always commendable to find experts who, with their suggestions and comments, help to improve the research work of their colleagues and the way it is disseminated.

We have tried to review in detail every aspect that you have commented on and have incorporated in the article the corrections you have made.

We hope the revised version is now suitable for publication.

Sincerely,

Elena Sandri, corresponding author.

[email protected]

1.Some deficiencies and areas for improvement in the Introduction section could be identified as follows:

(1) The introduction discusses the general importance of beverage consumption and its impacts; however, it does not clearly highlight the specific research gaps that this study intends to address.

Thank you very much for your suggestion, we have corrected it by adding to the introduction a part where we have made a bibliographical review analysing the gaps of study that this research comes to cover.

(2) The introduction covers a wide range of beverage types, related health impacts, socio-demographic factors, and lifestyle variables. Narrowing the focus to a specific type of beverage or health impact may make this study more scientifically rigorous and manageable. (3) The introduction could be shorter and more concise, focusing on key aspects that pertain directly to the study. Excessive detail or broad generalizations can dilute the main points.

Thank you for your suggestion, we have tried to shorten the introduction by eliminating some examples or more descriptive parts which, as you indicate, could distract from the objective of the study. 

We can remove types of beverages, but it seems to us that this would mean losing some of the information collected in the questionnaire and statistical power. We are only looking at five types of beverages in the study: water, sugar-sweetened beverages, juice, coffee and energy drinks and alcohol.   

(4) The Introduction lacks a clear statement regarding the research objectives. It is not clear what the specific predictions or expectations of the research are.

Thank you for your suggestion, we have reworded the paragraph explaining the objectives of the present study to make them clearer and more visible.

(5) The introduction does not provide a comprehensive review of existing literature on this subject. Although many references have been cited, there is no clear overview of previous findings or theories that could help contextualize the current research.

As indicated in point 1, we have added a literature review to help contextualise this research. 

 2.The Materials and Methods section of this study has several deficiencies and areas that could be improved.

(1) By limiting the sample to those with internet and social network access, and between the ages of 18 and 45, the study introduces a potential sampling bias that may limit the generalizability of results to wider populations. (2) The use of non-probabilistic snowball sampling may lead to biased results because the sample may not adequately represent the population. Additionally, the use of social networks or personal contacts may introduce selection bias. (3) The use of a self-developed questionnaire can be a potential source of bias as it may lack standardized metrics or may not have been validated in diverse populations.

We agree that the study, like any other study, has its strengths and weaknesses represented in some biases. At the suggestion of reviewer 1, we have expanded the section on the strengths and weaknesses of the study by setting out in detail all the possible biases it has, as well as detailing all the means that have been employed to minimise their impact.

 3.The Discussion section of this study had several limitations. The following are some key points to consider:

(1) The discussion suggests the design and dissemination of public health actions based on some of the socio-demographic variables studied. While these results provide a useful starting point, it is important to not overextend the implications of the findings.

Thanks for the suggestion, we have added a few paragraphs to the discussion where we point to the need for further specific studies to support the results that this research seems to point to.

(2) While the authors do refer to previous related studies, they could offer more direct comparisons between their own findings and the results of these previous studies to solidify their results, arguments, and conclusions. (3) Certain aspects, such as local economic conditions, educational facilities, and public health campaigns might have influenced both variables studied (beverage consumption and sociodemographic factors), potentially confounding the results.

This is certainly a possibility given that food consumption habits and lifestyles are complex and multifaceted realities influenced by a multitude of very different factors (tastes, habits, social conditioning factors, purchasing power, to name but a few). In order to give more solidity to the results obtained, it has been ensured that the sample was very large and that different statistical tests were used in order to analyse the results from different perspectives.

(4) While the authors do note the use of an online survey as a limitation, the discussion does not elaborate on other potential limitations such as the potential for recall bias, self-selection bias, non-response bias, or limitations inherent to cross-sectional studies.

As explained above, we have expanded the strengths and weaknesses section of the study by outlining all possible biases and detailing all the means employed to minimise their impact.          

(5) The discussion proposes a number of interesting directions for future research but does not provide hypotheses or methods on how this research should be conducted.

While we understand that a detailed description of the study design of this new research is beyond the scope of this article, we have added a few paragraphs in an attempt to further detail how this future research might be approached.

4.This conclusion restates the essential findings of this study. Here, the conclusion seems to summarize only a fraction of the study's full results.

We have expanded the conclusions section by incorporating further conclusions that can be drawn from the results obtained, as suggested.

Reviewer 3 Report

Comments and Suggestions for Authors

From the first of the manuscript screening by Turnitin the similarity index was 23% (still tolerable), but there is similarity 13% from single source with the title " The Relationship between Beverages Consumption and Cognitive Impairment in Middle-Aged and Elderly Chinese Population by Xinting Jiang 1,2,† [ORCID] , Liang Cui 1,†, Lin Huang 1, Yihan Guo 3 [ORCID] , Gaozhong Huang 2,* and Qihao Guo 1,* and also from MDPI"

Please revised strictly and resubmit for review

Comments on the Quality of English Language

From the first of the manuscript screening by Turnitin the similarity index was 23% (still tolerable), but there is similarity 13% from single source with the title " The Relationship between Beverages Consumption and Cognitive Impairment in Middle-Aged and Elderly Chinese Population by Xinting Jiang 1,2,† [ORCID] , Liang Cui 1,†, Lin Huang 1, Yihan Guo 3 [ORCID] , Gaozhong Huang 2,* and Qihao Guo 1,* and also from MDPI"

Please revised strictly and resubmit for review

Author Response

Dear Reviewer 3

Thank you very much for taking the time to review our article. We have revised the text of the article in detail and reworded some paragraphs to remedy the suggested copy rate.

Sincerely,

Elena Sandri, corresponding author.

[email protected]

Round 2

Reviewer 2 Report

Comments and Suggestions for Authors

The text has been revised by incorporating feedback provided by the reviewer. The authors have addressed my comments and suggestions, thereby enhancing the quality and clarity of the manuscript. I am now in a position to recommend acceptance.